# Deriving tropospheric ozone from assimilated profiles

Jacob C. A. van Peet[1,2] and Ronald J. van der A[1,3]

[1]Royal Netherlands Meteorological Institute (KNMI), De Bilt, The Netherlands
[2]Vrije Universiteit Amsterdam, Department of Earth Sciences, Amsterdam, The Netherlands
[3]Nanjing University of Information Science & Technology (NUIST), Nanjing, China

**Correspondence:** van Peet (j.c.a.van.peet@vu.nl), van der A (avander@knmi.nl)

**Abstract.** We derived global tropospheric ozone ($O_3$) columns from GOME-2A and OMI $O_3$ profiles, that were simultaneously assimilated into the TM5 global chemistry transport model for the year 2008. The horizontal model resolution has been increased by a factor of 6 for more accurate results. To reduce computational cost, the number of model layers has been reduced from 44 to 31. The model ozone fields are used to derive tropospheric ozone, which is defined here as the partial column
between mean sea level and $6\,\mathrm{km}$ altitude. Two methods for calculating the tropospheric columns from the free model run and assimilated $O_3$ fields are compared. In the first method, we calculate the residual between assimilated total columns and the partial model column between $6\,\mathrm{km}$ and the top of atmosphere. In the second method, we perform a direct integration of the assimilated $O_3$ fields between the surface and $6\,\mathrm{km}$. The results are validated against tropospheric columns derived from ozone sonde measurements. Our results show that the residual method has a too large variation to be used reliably for the
determination of tropospheric ozone, so the direct integration method has been used instead. The median global bias is smaller for the assimilated $O_3$ fields than for the free model run, but the large variation makes it difficult to make definitive statements on a regional or local scale. The monthly mean ozone fields show significant improvements and more detail when comparing the assimilated $O_3$ fields with the free model run, especially for features such as biomass burning enhanced $O_3$ concentrations and outflow of $O_3$ rich air from Asia over the Pacific.

## 1 Introduction

Tropospheric ozone has direct and detrimental effects on human health (Beck et al., 1998; WHO, 2013). It mostly affects the respiratory tract and the lungs, causing e.g. shortness of breath, coughing and a reduced lung function. Respiratory illnesses such as asthma and bronchitis are aggravated by exposure to ozone. Long-term exposure to ozone might increase the mortality rate due to respiratory illnesses. Ozone also negatively affects ecosystems and crop yield because it reduces photosynthesis and
plant growth (EPA, 2013). Because plants react differently to exposure to ozone, the balance between species in an ecosystem may shift as well. Monks et al. (2015) give an extensive review on tropospheric ozone and its precursors in relation to air quality and climate.

Apart from the direct and indirect effects on living organisms, ozone is also a greenhouse gas. It strongly absorbs solar radiation below $300\,\mathrm{nm}$, which is why the temperature of the stratosphere is increasing with altitude. Therefore, understanding
the ozone distribution is important for understanding the thermal structure of the atmosphere.

Ozone occurs naturally in the troposphere, but concentrations have increased due to human activity. Locally, ozone is produced primarily by reaction cycles involving carbon monoxide, NOx, methane and other hydrocarbons. The most important source sectors of these pollutants are transport and industry. Photodissociation of tropospheric ozone is the main source of OH, which has a major role in removing pollutants from the atmosphere. Ozone can also be transported from the stratosphere down to the troposphere in stratosphere-troposphere exchange events.

The tropospheric ozone column is defined as the total ozone amount per unit area between the surface and the tropopause. However, near the tropopause, stratosphere-troposphere exchange of air may occur, which can lead to an under- or over-estimation of the lower tropospheric ozone column. Since the tropospheric ozone in the lower layers has the highest impact on living organisms, we will focus on the partial ozone column between the surface and $6\,km$ above mean sea level. Satellite measurements are not very sensitive close to the surface, but in the chosen altitude range some information from the measurements is still present (see Figure 4). Because the top level is at a fixed altitude, it will be referred to as the fixed altitude top level (FAT) hereafter. The corresponding 0–$6\,km$ ozone partial column will be referred to as the FAT column.

Tropospheric ozone can be determined by a number of satellite based methods. In nadir-limb matching techniques, the integrated profile from a limb instrument is subtracted from the total column for the same air mass. Limb profiles and total columns can be obtained from the same instrument (e.g. SCIAMACHY; van der A, 2001; Ebojie et al., 2014), but also from different instruments on the same satellite (e.g. OMI total column and MLS limb profile; Ziemke et al., 2006). In Schoeberl et al. (2007), the horizontal resolution of the MLS limb profiles was increased by trajectory calculations before subtracting them from the OMI total columns. Tropospheric ozone columns were also derived from assimilated OMI total columns and MLS limb profiles by Stajner et al. (2008). Using only nadir observations, Fishman and Balok (1999); Fishman et al. (2003) combined Total Ozone Mapping Spectrometer (TOMS) total columns and Solar Backscattered Ultraviolet (SBUV) stratospheric profiles and determined tropospheric ozone with the empirically corrected tropospheric ozone residual method. Assimilated GOME profiles were subtracted from GOME/TOMS total columns by de Laat et al. (2009).

The methods mentioned above all use the UV-VIS range of the spectrum. There are also a number of ozone emission lines in the thermal infrared (i.e. the wavelength range where the atmosphere emits radiation, instead of reflecting solar light), most notably near $9.6\,\mu m$. This emission line can also be used by satellite instruments (e.g. IASI) to measure ozone (e.g. Boynard et al., 2018).

In the tropics, the cloud top height is very stable at an altitude of approximately $200\,hPa$. Therefore, cloudy scenes can be used to obtain the above-cloud ozone column while the cloud-free scenes can be used to obtain total ozone columns. The difference between these two values is the ozone column below the cloud top. This convective-cloud-differential method (Ziemke et al., 1998) has recently been applied to European satellite measurements to study the trends in a $20\,year$ time series and as a preparation for the TROPOMI mission (Heue et al., 2016).

Outside the tropics, the cloud top height varies too much to reliably obtain ozone columns using the convective-cloud-differential method. UV-VIS retrievals are not very sensitive to the altitude where tropospheric ozone is located, so direct integration of UV-VIS ozone profiles does not provide a viable alternative either. The height information can be restored by using data assimilation, where information from ozone profiles, averaging kernels and the chemical transport model are combined.

The sensitivity and information content of UV-VIS retrievals is higher in the stratosphere, therefore an alternative approach is to subtract stratospheric columns, derived from assimilated ozone profiles, from accurate total columns (for example from DOAS retrievals). The remainder is taken as the residual tropospheric column (de Laat et al., 2009).

The assimilation of ozone measurements from satellites is usually done by either 4DVAR or (ensemble) Kalman filters. For example, 4DVAR data assimilation has been used for the ERA-Interim dataset (Dragani, 2011) and in the Belgian Assimilation System for Chemical ObsErvations (BASCOE, Errera et al. (2008)). The stratospheric ozone analyses from the BASCOE system have been evaluated by Lefever et al. (2015), and it has been coupled to the Integrated Forecast system of the ECMWF (Huijnen et al., 2016). Ensemble Kalman filters (Evensen, 2003; Houtekamer and Zhang, 2016) have been used for the assimilation of multiple trace gas measurements by Miyazaki et al. (2012). In this research, we use the Kalman filter as described in Segers et al. (2005); van Peet et al. (2018) for the assimilation of ozone profiles from the GOME-2 and OMI UV-VIS satellite instruments.

The assimilated ozone fields will be used to derive tropospheric columns in two ways. One method is to integrate the assimilated ozone column up to the FAT, hereafter called the FAT column (i.e. the column between the surface and $6\,\mathrm{km}$ altitude). The other method is to take the difference between the integrated assimilated profile from the FAT to the top of the atmosphere and the assimilated total ozone columns from the Multi Sensor Reanalysis (MSR; van der A et al., 2010, 2015), hereafter called the residual-FAT column. The MSR is a long-term (1970–2017) data set of assimilated total columns from all available satellite measurements.

## 2 Methodology

We use the ozone profiles from the UV-VIS instruments GOME-2 (Callies et al., 2000; Munro et al., 2016) and OMI (Levelt et al., 2006) that are described in van Peet et al. (2018). The ozone profiles from both instruments are retrieved with the optimal estimation technique. For GOME-2 the algorithm is described in van Peet et al. (2014), while the OMI algorithm is described in Kroon et al. (2011). The ozone profiles are assimilated into the global chemistry transport model TM5 (Tracer Model, version 5; e.g. Krol et al., 2005). Two major changes with respect to the settings used in van Peet et al. (2018) are an increased model resolution and a change from operational to ERA-Interim (Dee et al., 2011) meteorological fields that drive TM5. The ERA5 reanalysis data was not yet available for use in the TM5-version used in the assimilation. Above $230\,\mathrm{hPa}$, the TM5 version in this research uses the parameterised ozone chemistry scheme version $2.1$ of Cariolle and Déqué (1986); Cariolle and Teyssèdre (2007). Below the $230\,\mathrm{hPa}$, the ozone concentrations are nudged towards climatological values.

To get more accurate assimilated ozone fields, the horizontal resolution of TM5 is increased from $3° × 2°$ to $1° × 1°$ (longitude × latitude). At the same time, the vertical resolution is decreased from $44$ to $31$ layers to reduce the computational cost. The original $44$ layers are a subset from the vertical grid used by the European Centre for Medium-Range Weather Forecasts (ECWMF) operational data stream, while the new $31$ layers are a subset from the vertical grid used for the ERA-Interim reanalysis. Below about $73\,\mathrm{hPa}$ ($19\,\mathrm{km}$), the layers are between $0.8$ and $1.5\,\mathrm{km}$ thick, until about $54\,\mathrm{km}$ every other level is selected and the layer thickness increases from $3$ to $5.5\mathrm{km}$, and the top four levels are all selected. It is not expected that the reduction

in vertical resolution affects the accuracy of the outcome, since the thickness of the model layers is still less than the estimated vertical sensitivity of the retrievals, which is about $7$–$10\,\mathrm{km}$ in the stratosphere (Hoogen et al., 1999; Liu et al., 2010).

The sensitivity of the retrieval to the true state of the atmosphere is given by the averaging kernel (AK). The trace of this matrix equals the Degree of Freedom for the Signal (DFS). The rows of the AK give an indication how the true profile is smoothed over the layers of the retrieval. An extended discussion on the information content that can be derived from AKs from GOME, SCIAMACHY, GOME-2 and OMI is presented by Keppens et al. (2018). AKs are also an important factor in intercomparison of different ozone retrieval algorithms such as in Meijer et al. (2006).

Like Segers et al. (2005), we assume that the spatial correlation between any two points in the 3D ozone field is constant in time, and that changes over time occur in the ozone standard deviation only. Therefore, the model covariance matrix is parameterised into a time independent correlation field and a time dependent uncertainty field. Due to the changes in resolution and meteorological fields, the correlation field had to be derived again according the same method as described in van Peet et al. (2018). No other changes have been made to the assimilation algorithm.

Since the horizontal resolution of the chemical transport model has been increased, the computational cost of the assimilation algorithm did also increase. In order to limit the total processing time only ozone profiles for the year 2008 were assimilated. TM5 was used in two runs: a free model run without assimilation of observations, and an assimilation model run with the simultaneous assimilation of both GOME-2 and OMI ozone profiles. For each model run, the FAT column was calculated by direct integration of the $O_3$ fields, and the residual-FAT column was calculated using the Multi Sensor Reanalysis (MSR, van der A et al., 2010, 2015) total columns. The total columns are distributed over the layers of the model proportionally to the subcolumn of that layer. The MSR-model uses the same parameterised ozone chemistry as the profile assimilation used in this research (Cariolle and Déqué, 1986; Cariolle and Teyssèdre, 2007), but with the more up-to-date version 2.9 of the chemistry parameters.

The results are validated against ozone sondes downloaded from the public World Ozone and Ultraviolet Radiation Data Center (WOUDC, WMO/GAW, 2016) database. Since the model produces $O_3$ fields with a $6\,\mathrm{h}$ interval at $0, 6, 12$ and $18\,\mathrm{hours}$ UTC, the maximum difference between sonde launch and model field time is set to three hours. The sonde profile is compared to the model profile from the gridcell containing the sonde launch site, no interpolation of the model field to the sonde launch location is performed. In order for the ozone sondes to be used in the validation, it should have reached a minimum altitude of $10\,\mathrm{hPa}$, and the integrated ozone profile should be between $100$ and $550\,\mathrm{DU}$.

## 3 Results

Figure 1 shows the monthly mean FAT columns for the year 2008. In general, the free model shows higher ozone concentrations than the assimilated ozone fields. The ozone chemistry parameterisation used in TM5 is known to overestimate ozone concentrations (Cariolle and Déqué, 1986; Cariolle and Teyssèdre, 2007), resulting in the higher ozone concentrations in the free model.

Note that since the FAT has a fixed altitude with respect to sea level, elevated regions such as Antarctica or the Tibetan Plateau show a small tropospheric column. The Northern hemisphere has a higher FAT column than the Southern hemisphere, and a yearly cycle can be clearly seen in the plots. The high ozone concentrations in the Northern Hemisphere have various sources such as stratosphere-troposphere exchanges and anthropogenic precursor emissions (Ziemke et al., 2011). An increase in ozone concentration is seen in the Southern Atlantic ocean for September, and between Africa and Australia in a zonal band around $-25°$ latitude. This increase can be attributed to biomass burning, and coincides with the month of maximum $NO_x$ concentration (an ozone precursor) in Africa (van der A et al., 2008). From March to September, transport of ozone rich air can be seen from Asia across the Pacific. Similar features in the yearly cycle of ozone are also observed in the tropospheric ozone climatology by Ziemke et al. (2011). This climatology is based on the residual of OMI total columns and MLS stratospheric columns (using the thermal tropopause definition), on a horizontal resolution of $5° × 5°$. Two sharply defined narrow zonal features of elevated ozone concentrations can be seen at $10°$ and $-20°$ latitude. These zonal features are also present in the free model run (left column in Figure 1), so they are not caused by the observations. Since the monthly mean (surface) pressure fields do not show a similar feature, it is unlikely that it is caused by the meteorological data that is used to drive the model. The most likely cause for these narrow zonal elevated ozone concentrations is therefore a model artefact. It should be noted that the difference is only a few DU, so these zonal features are not easily observed in total column maps.

In the assimilation, the spatial correlation is assumed to be constant. The correlation is derived by the same method as described in van Peet et al. (2018). The errors on the profile are part of the assimilation output, and combining these the correlation matrix and error profiles in a post-processing step allows the reconstruction of the covariance matrix in the vertical direction. The tropospheric part of this covariance matrix can be used to derive the relative error on the tropospheric column. In Figure 2 the yearly mean assimilated tropospheric FAT column is given in panel (a), while the associated relative errors are shown in panel (b). The yearly mean FAT columns varies between 3 and 23 DU, while the relative error varies between 11 and 18 %, depending on the location on Earth.

In order to estimate the impact of the upgraded TM5 model resolution and meteorological data used to drive TM5, we validate the resulting tropospheric ozone columns with ozone sondes (from the surface up to approximately 30 km). Figure 3 shows absolute and relative biases for both the free model run and assimilated $O_3$ fields. There is a significant improvement of the assimilated $O_3$ fields over the free model run when compared to ozone sondes, with the exception of the UTLS (around 15 km). The sharp ozone gradients in this altitude range are not captured fully by the model and the satellite observations. These results are comparable to the TM5 model run used in van Peet et al. (2018, see their figure 13), where the same satellite data was assimilated into TM5, running on a coarser model resolution and with operational meteo data. In van Peet et al. (2018), the median bias for the tropospheric column is between $-5$ to $0$ % for the period 2008–2011, while in the current research it is between $-2$ and $3$ % for 2008 only.

The assimilation of the satellite data has the largest impact in the altitude range between 100 and 5 hPa. This is also the region where the retrievals are most sensitive to the measurements. To illustrate this, we averaged the AKs of all GOME-2 and OMI measurements that were assimilated into the model. The resulting mean AKs are plotted in figure 4, with a marker at the altitude of the retrieval layer. The AKs have the largest value in the altitude range where the improvement of the of the

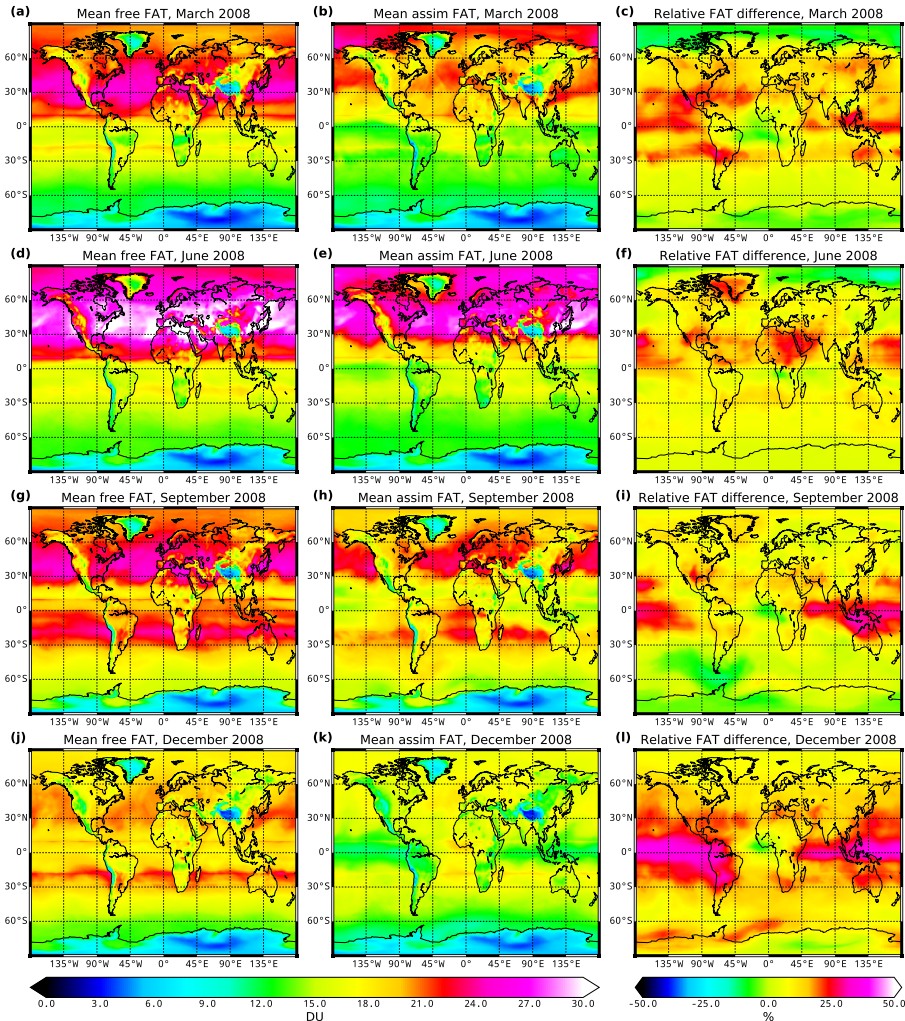

**Figure 1.** Monthly mean tropospheric O$_3$ fields. Left column (a, d, g, j): free model run, middle column (b, e, h, k): assimilated O$_3$ fields, right column: the relative difference ((free-assim)/free). From top to bottom: March (a, b, c), June (d, e, f), September (g, h, i) and December (j, k, l) 2008.

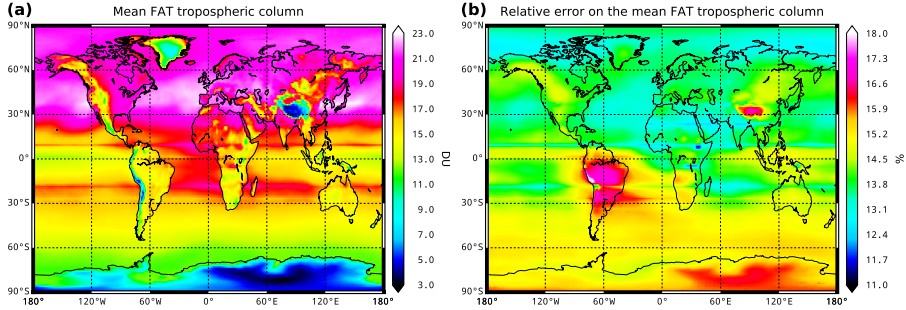

**Figure 2.** (a) Yearly mean FAT tropospheric column, derived from assimilated ozone fields, and (b) the corresponding relative error.

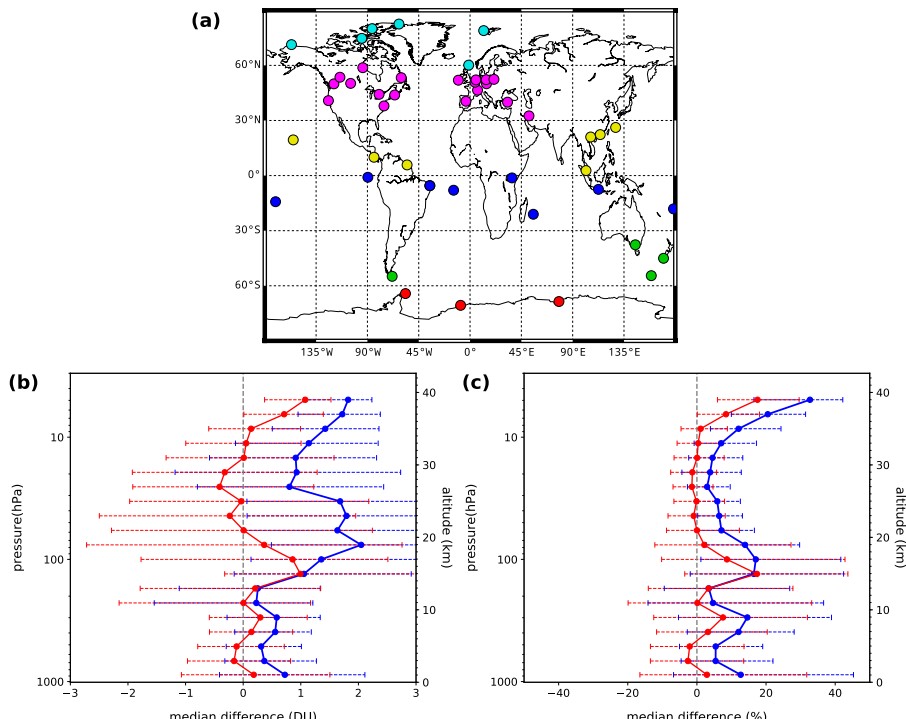

**Figure 3.** TM5 validation results with respect to sondes. The top plot (a) shows the locations of all sondes used in the validation of the model. The color coding of the sondes is the same as in Figure 5. Bottom left (b): median absolute difference, bottom right (c): median relative difference. The blue line is the model run without assimilation, the red line is the model run with assimilation of GOME-2 and OMI. The error bars indicate the range between the 25th and 75th percentile.

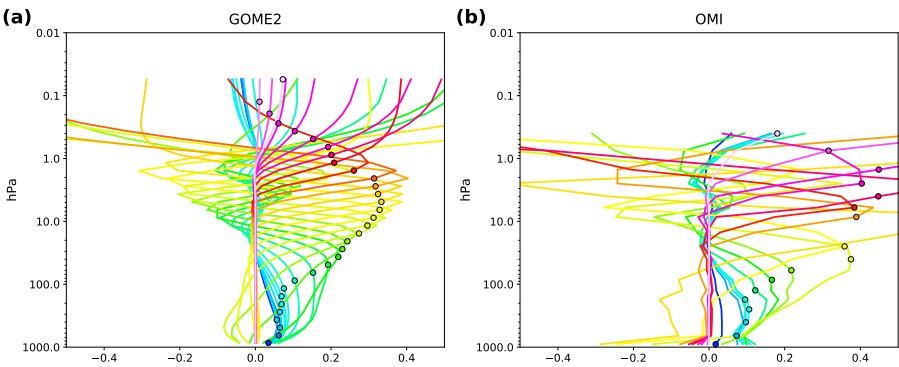

**Figure 4.** Mean AKs calculated from all assimilated retrievals from GOME-2 (left) and OMI (right). The markers show the altitude of the retrieval layer.

assimilated $O_3$ fields over the free model run are the largest. The oscillations in the AKs near the top of the retrieval have limited effect because the $O_3$ concentration is small in that altitude range.

In Figure 5, scatterplots of the FAT columns are shown for the free model run and the assimilated $O_3$ fields and of the residual-FAT column for the assimilated $O_3$ fields only. The data are grouped according to ozone sonde station location. The free model run and assimilated $O_3$ fields perform comparably and both have a higher correlation coefficient than the residual method (see Figure 5d). The residual method shows some negative columns, indicating that the stratospheric part of the assimilated profiles is larger than the total column from the MSR. Residual-FAT columns based on the free model run show even more negative values, so they are not shown in the figure. The residual method has a lower correlation coefficient and a higher uncertainty than the FAT columns of the free model run and assimilated $O_3$ fields, and therefore will be omitted from the subsequent analysis.

We can see from Figure 3 that the bias with respect to sondes in the troposphere is smaller for the assimilated $O_3$ fields than for the free model run. Figure 5 shows that the root mean square ($\mathrm{rms}$) and correlation for the assimilated $O_3$ fields slightly improve compared to the free model run. To further investigate the variation between TM5 model results and sonde measurements, the $\mathrm{rms}$ and $\mathrm{mean}$ differences between the model and sonde FAT columns are plotted in Figure 6. The figure gives the RMS for all collocations (with a minimum of 10) per station as a function of latitude on the top left, and the mean difference on the top right. The green dots in the maps indicate stations where the absolute value of the $\mathrm{rms}$ (or $\mathrm{mean}$) from the assimilated $O_3$ fields is smaller than for the free model run ($|\mathrm{rms}_{\mathrm{assim}}| < |\mathrm{rms}_{\mathrm{free}}|$ or $|\mathrm{mean}_{\mathrm{assim}}| < |\mathrm{mean}_{\mathrm{free}}|$). The red dots indicate stations where the reverse is true ($|\mathrm{rms}_{\mathrm{assim}}| > |\mathrm{rms}_{\mathrm{free}}|$ or $|\mathrm{mean}_{\mathrm{assim}}| > |\mathrm{mean}_{\mathrm{free}}|$ ). In the southern hemisphere (lat $< -30$), the assimilated $O_3$ fields show a smaller $\mathrm{rms}$ and a smaller absolute value of the $\mathrm{mean}$ for $4$ and $5$ out of $7$ stations, respectively. In the tropics ($-30 \leq \mathrm{lat} < 30$), the assimilated $O_3$ fields show a smaller $\mathrm{rms}$ and a smaller absolute value of the $\mathrm{mean}$ for $9$ and $10$ out of $14$ stations, respectively. The assimilated $O_3$ fields perform better than the free model run for the majority of the tropical stations, but note that the $\mathrm{rms}$ and the absolute value of the $\mathrm{mean}$ are larger than at higher

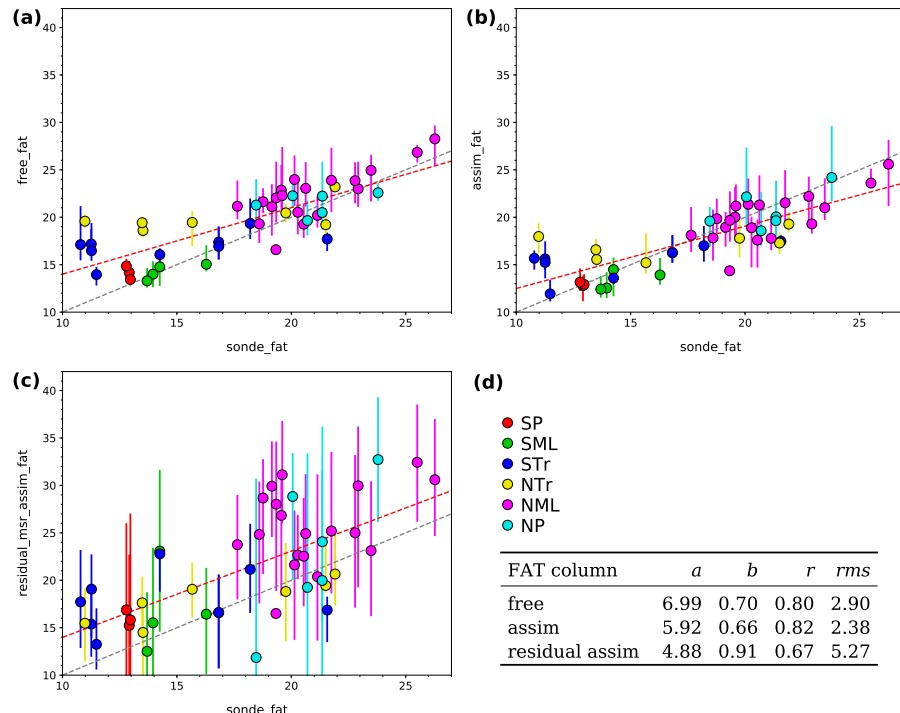

**Figure 5.** Scatterplots of tropospheric columns based on model output versus sonde measurements. The plot symbols are the median values of collocations grouped by station. The error bars indicate the 25–75 percentiles of the distribution. Top (a): free model run, top right (b): assimilated $O_3$ fields, bottom left (c): residual-FAT column for the assimilated $O_3$ fields. Colors indicate $30°$ latitude bands: SP = South Pole, SML = Southern MidLatitudes, STr = Southern TRopics, NTr = Northern TRopics, NML = Northern MidLatitudes, NP = North Pole. The grey dashed line is the 1:1-line and the red dashed line gives the best linear fit to the data. The fit parameters are listed in the table at the bottom right (d). The columns marked $a$ and $b$ are the linear fit parameters of the line $a + bx$, $r$ is the linear Pearson correlation coefficient, $rms$ is the root mean square between the values on both axes. The number of stations included in each plot is $48$.

latitudes. In the northern hemisphere ($\text{lat} \geq 30$), the assimilated $O_3$ fields show a smaller $\text{rms}$ and a smaller absolute value of the $\text{mean}$ for 9 and 13 out of 24 stations, respectively.

To study temporal variation, time series of monthly median FAT-columns are shown in Figure 7 for three different latitude bands. For the northern hemisphere (Figure 7, panel (a)), the free model run is closer to the sondes than the assimilated $O_3$ fields for January till May. The assimilated $O_3$ fields are closer to the sonde measurements than the free model run from June till December. For the lapse rate tropopause (not shown here), the assimilated $O_3$ fields are closer to the sonde data than the free model run throughout the year. Since in the troposphere the model is nudged towards an ozone climatology (Fortuin and Kelder, 1998), the climatological value for each collocation has been calculated and the monthly median is also shown in Figure 7. The free model run follows a similar pattern as the climatological values. It should be noted that the free model

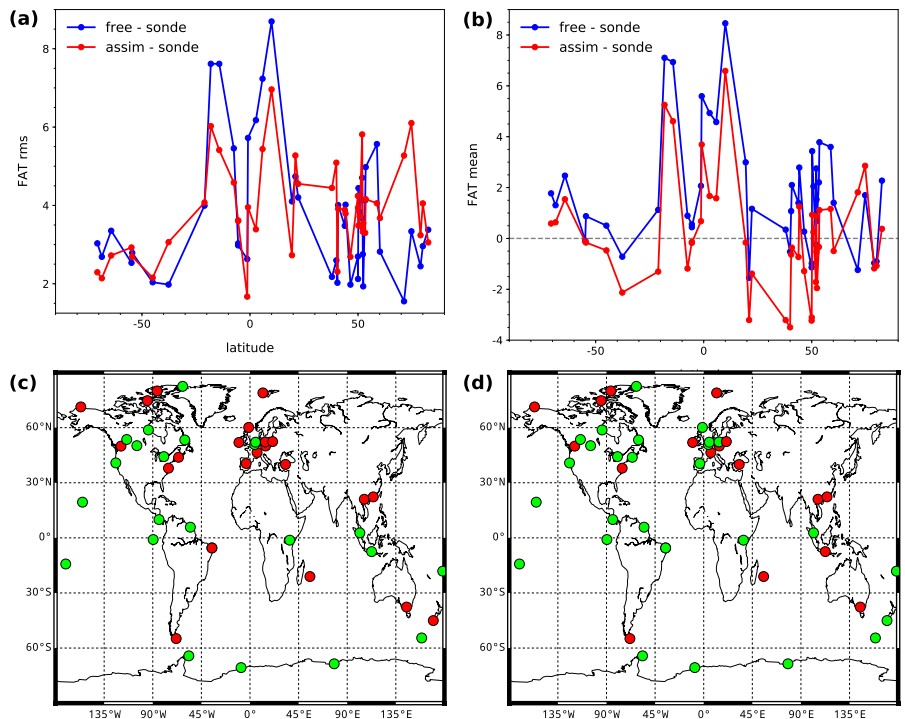

**Figure 6.** The FAT rms (a) and mean (b) per station as a function of latitude. The blue line gives the results for the free model run compared to sondes. The red line gives the results for the assimilated $O_3$ fields compared to sondes. Bottom left (c): green dots indicate stations where $|\mathrm{rms_{assim}}| < |\mathrm{rms_{free}}|$, red dots where $|\mathrm{rms_{assim}}| > |\mathrm{rms_{free}}|$. Bottom right (d): green dots indicate stations where $|\mathrm{mean_{assim}}| < |\mathrm{mean_{free}}|$, red dots where $|\mathrm{mean_{assim}}| > |\mathrm{mean_{free}}|$. Only results for stations with at least 10 collocations have been plotted.

run and assimilated $O_3$ fields start with the same ozone concentrations. Due to the assimilation of observations they diverge quickly, and the monthly median values for January are not the same.

For the tropics (Figure 7, panel (b)), the ozone sonde FAT columns are lower than the assimilation model run, which in turn is lower than the free model run throughout the year. This behavior is consistent with the plots shown in Figure 6. For the southern hemisphere (Figure 7, panel (c)), both model runs, sonde measurements and climatological values are close together, except for November and December, which might be a consequence of the ozone hole. Note that for all three latitude bands, the differences are very small, in the order of 2–3 DU, and close to the uncertainty.

As an example of the FAT-column variability throughout the year, Figure 8 shows time series for the free model run and assimilated $O_3$ fields, and for the sonde measurements over three different stations: the Antarctic station Neumayer ($8.26°$ W, $70.56°$ S), the tropical station Hilo ($155.04°$ W, $19.43°$ N) and the northern hemisphere station Lerwick ($1.19°$ W, $60.14°$ N). Time series for all sonde station locations used in this research with more than 10 collocations with the model output are available in the supplementary material online. For the Neumayer station, the free model run and assimilated $O_3$ fields give comparable results during the polar night. The decrease in the tropospheric column that is visible from October onward is

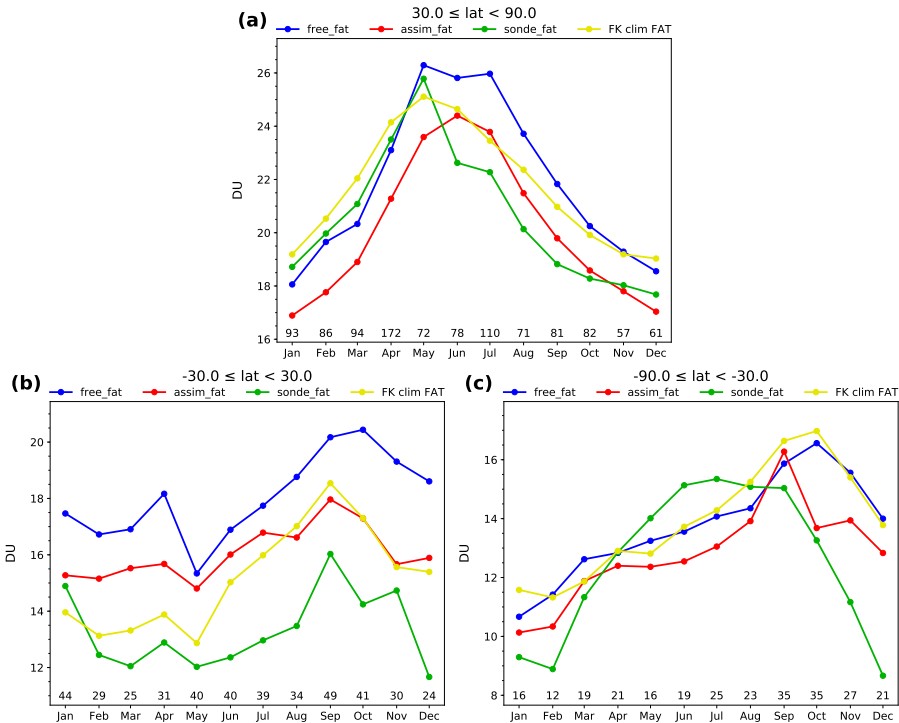

**Figure 7.** Time series of monthly median global FAT-columns for (a) southern hemisphere ($-90 \leq \mathrm{lat} < -30$), (b) tropics ($-30 \leq \mathrm{lat} < 30$) and (c) northern hemisphere ($\mathrm{lat} \geq 30$). Blue line: free model run, red line: assimilated $O_3$ fields, green line: sonde data, yellow line: Fortuin and Kelder climatology. The numbers along the x-axis indicate the number of collocations between model and sondes. Note the different scale on the y-axis in each panel.

caused by solar radiation and $NO_x$ induced $O_3$ destruction, not by the halogen induced destruction of the ozone hole (see e.g. Helmig et al., 2007). For the Hilo station, the assimilated $O_3$ fields shows systematically lower FAT columns than the free model run. The FAT columns from the assimilated $O_3$ fields are in better agreement with the sonde FAT columns than the free model run. For the Lerwick station, the free model run and assimilated $O_3$ fields show similar FAT columns, and the rms bias of the assimilated $O_3$ fields is larger than for the free model run. However, the absolute value of the mean bias is larger for the free model run than for the assimilated $O_3$ fields.

## 4   Discussion

Deriving tropospheric ozone from nadir looking UV-VIS instruments is a big challenge due to the limited sensitivity of these instruments in the troposphere. Since most of the radiation in the wavelength range between $280$ and $330\,\mathrm{nm}$ is absorbed by the ozone layer, only a small part reaches the surface. Typical values for the Degree of Freedom for Signal (DFS, a measure

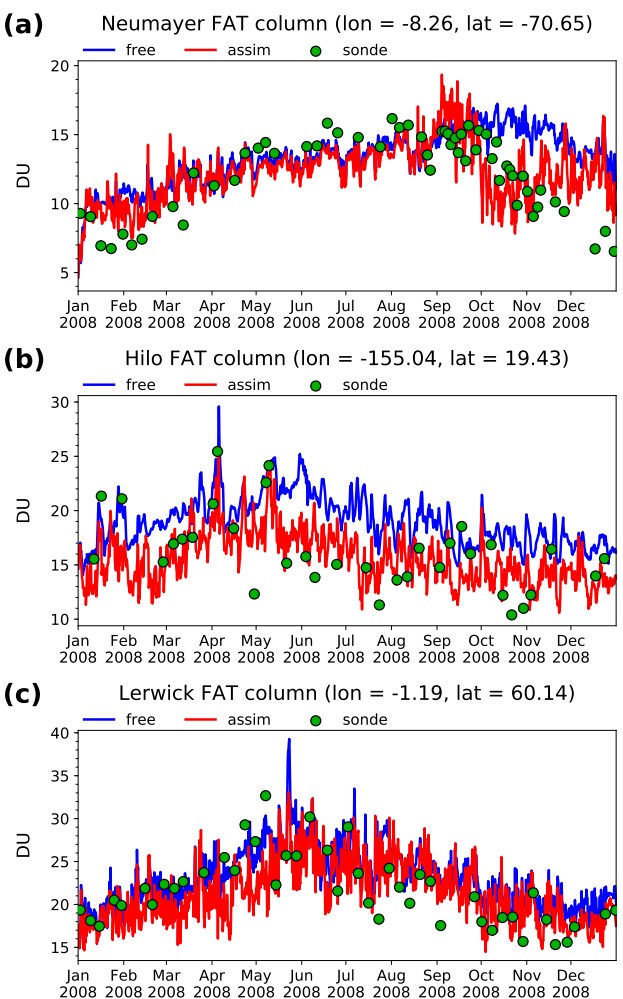

**Figure 8.** Three time series of collocated model output and ozone sonde measurements. From top to bottom: Neumayer (a), Hilo (b) and Lerwick (c). The station coordinates have been indicated in the plot titles. Blue line: FAT-column from model run without assimilation, red line: FAT-column from model run with assimilation of GOME-2 and OMI measurements, green circles: ozone sonde measurements.

for the number of independent pieces of information in the retrieval) of the tropospheric column are between $\sim 0.5$ at higher latitudes to $\sim 1.2$ in the tropics (Liu et al., 2005).

Both the DOAS total columns used in the MSR and the UV-VIS stratospheric partial columns from the retrievals used in this research are accurate measurements of the ozone concentration. The large variation in the residual-FAT column was therefore unexpected and we discuss the differences between both assimilation systems in some more detail. The MSR only assimilates total columns, which are distributed over the layers of the model proportionally to the subcolumn of that layer. The MSR-model uses the same parameterised ozone chemistry as the profile assimilation used in this research (Cariolle and Déqué, 1986; Cariolle and Teyssèdre, 2007), but with a more up-to-date version of the chemistry parameters (2.9 for the MSR, 2.1 for this research). However, since both assimilation systems are frequently updated with observations, it seems unlikely that the difference in parameterisation version plays a major role in the observed residual-FAT column variation. Also, data from all available total ozone satellite sensors is assimilated into the MSR instead of only the profiles from the two GOME-2 and OMI instruments that are assimilated into the current system. The observations are both bias corrected, the total columns with respect to Brewer-Dobson measurements and the profiles with respect to sondes. The MSR-model resolution is $0.5° \times 0.5°$, while the profile assimilation runs on $1° \times 1°$. The most extreme negative residuals are found for the Antarctic sonde stations, so high solar zenith angles may have some effect. However, since negative residuals are also found at lower latitudes, it cannot be the only explanation.

Since the residual-FAT column cannot be used reliably for determining the tropospheric ozone column, the directly integrated FAT columns from the assimilated $O_3$ fields might offer an alternative. The global median difference with $O_3$ sondes is clearly lower for the assimilated $O_3$ fields than for the free model run (see Figure 3). However, this is not so clear from the scatterplots of the FAT columns grouped by station (see Figure 5). The spatial distribution is also much better for the assimilated $O_3$ fields than for the free model run (see Figure 1). This can be seen in, for example, the outflow of ozone rich air from Asia over the Pacific and biomass burning enhanced $O_3$ concentrations.

There are several potential explanations for the small improvements of the assimilation tropospheric ozone columns compared to the free model run. The reduced sensitivity in the troposphere of GOME-2 and OMI is compensated for by incorporating the averaging kernel into the observation operator, and the tropospheric column is changed due to the assimilation. However, the tropospheric uncertainties of the observations might be too large to reduce the model uncertainties, so the improvement due to the assimilation only becomes clear when looking at the global median results.

The parameterised chemistry version that is being used is known to overestimate low latitude ozone in the troposphere (Cariolle and Teyssèdre, 2007). Below $230 \, \mathrm{hPa}$ however, the model is nudged towards the climatology of Fortuin and Kelder (1998). Above $230 \, \mathrm{hPa}$ the full Cariolle chemistry scheme is used, but two of the parameters in that scheme (i.e. the average volume mixing ratio and the overhead ozone column) are set to the climatological values.

Other possible factors contributing to the large variation in the FAT columns are the representation errors between the model and sondes and between model and observations. Since TM5 is running on a $1° \times 1°$ horizontal grid, the model ozone concentrations are an average over the grid cell while the ozone sonde measurements are point sources. In mountainous regions, the altitude of the model grid cell might also not correspond to the altitude of the sonde station. The ground pixel size and

location of the satellite observations might not coincide with the model grid cells either. For example, the footprint size of the GOME-2 measurements used in this research is about $160 \times 160$ km, which is larger than the model grid cells. The satellite instruments ground pixel centre determines in which model grid cell the pixel is assimilated.

Throughout the year, the FAT column from the assimilated $O_3$ fields is smaller than the FAT column from the free model run (Figure 7). This is consistent with the validation results for the whole profile (Figure 3), and with the rms values between model and sondes in the scatterplots of Figure 5. The northern hemisphere sonde FAT columns are closer to the free model run from January till May, but closer to the assimilated $O_3$ fields from June till December. The reason for the "smallest bias" shift from the free model run to the assimilated $O_3$ fields is unknown, but it should be stressed that the differences are small (in the order of 2–3 DU) and close to the uncertainty. If, instead of the FAT column, the column based on the lapse rate tropopause is used, such a "smallest bias" shift does not occur and the bias with respect to the assimilated $O_3$ fields run is always smaller than for the free model run.

## 5 Conclusions

Ozone profiles retrieved from GOME-2A and OMI measurements were assimilated simultaneously into the TM5 global chemistry transport model for the year 2008. With respect to the model version used in van Peet et al. (2018), the horizontal resolution of TM5 is increased from $3° \times 2°$ to $1° \times 1°$ (longitude $\times$ latitude). At the same time, the vertical resolution is decreased from 44 to 31 layers to reduce the computational cost. The meteorological data used to drive the model has also been upgraded from the operational data stream from the ECMWF to the ERA-Interim data set. Due to the large variation in the residual-FAT columns in the current model setup, they can't be used reliably, and the direct integrated FAT columns should be used instead. The median global bias with respect to $O_3$ sondes is smaller for the assimilated $O_3$ fields than for the free model run. When the tropospheric $O_3$ columns are grouped according to station, the root mean square of the median sonde columns and model output is smaller for the assimilated $O_3$ fields than for the free model run. The rms for each station separately also shows an improvement for the majority of stations on the southern hemisphere and in the tropics. The absolute value of the bias is also smaller for the assimilated $O_3$ fields than for the free model run for the majority of stations globally. The monthly median global FAT columns show a small bias with respect to ozone sonde measurements for the free model January till May, but from June till December, the assimilated $O_3$ fields have the smallest biases with respect to ozone sondes. The monthly mean ozone fields show significant improvements and more detail when comparing the assimilated $O_3$ fields with the free model run, especially for features such as biomass burning enhanced ozone concentrations and outflow of ozone rich air from Asia over the Pacific.

*Data availability.* OMI ozone profiles are operationally retrieved and can be obtained from NASA's Goddard Earth Sciences (GES) Data and Information Services Center (DISC) on-line archive at https://aura.gesdisc.eosdis.nasa.gov/data/Aura_OMI_Level2/OMO3PR.003/. GOME-2 ozone profiles are specifically retrieved for this research and can be obtained by contacting the author. Although not used in this research,

operationally retrieved GOME-2 ozone profiles can be retrieved from EUMETSATs ACSAF (https://acsaf.org/index.html), but note that a registration is required.

*Author contributions.* JvP has performed the calculations and analyses of the research and he wrote the paper with comments from RvdA. RvdA was involved in the conceptualization of the paper and had a supervising role.

5   *Competing interests.* The authors declare that they have no conflict of interest.

*Acknowledgements.* The authors acknowledge all scientists and institutes who contributed their ozone sonde data to the World Ozone and Ultraviolet Radiation Data Center (WOUDC, WMO/GAW, 2016), and the Meteorological Service of Canada for hosting this important public database. EUMETSAT is acknowledged for providing the GOME-2 L1 data and Olaf Tuinder and Robert van Versendaal for their help in the retrieval of the GOME-2 ozone profiles. The Dutch-Finnish OMI instrument is part of the NASA EOS Aura satellite payload. The
10   OMI ozone profiles (OMO3PR, v003) were retrieved at NASA Goddard Earth Sciences Data and Information Services Center (GES DISC) and accessed from the local storage at the Royal Netherlands Meteorological Institute (KNMI). Part of this research has been funded by the Ozone_cci project (http://www.esa-ozone-cci.org), which is part of the Climate Change Initiative (CCI) program of the European Space Agency (ESA).

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
