# Peer review of "Deriving tropospheric ozone from assimilated profiles"

_Atmospheric Chemistry and Physics, 2018_

## Referee Comment (RC1) · Anonymous Referee #1 · 28 Feb 2019

Deriving tropospheric ozone from assimilated profiles:

van Peet and van der A. aim at improving the TM5 simulation by assimilating OMI and GOME-2 ozone into the model. The results of the paper show the clear positive impact of the assimilation making it very interesting and worthy of being published. In general the figures are clear and well done but the syntax and the grammar can be improved. My main comments for the paper are that it lacks context; that is, it is not well discussed in context of what is done so far for ozone assimilation and how this work fits in the bigger picture. I recommend the publication of this paper after the following comments and suggestions are addressed:

General comments

[Figure]

- The averaging kernels of OMI and GOME2-A can help you understand and analyze your results (or at least a discussion of AKs e.g. the recent publication by Keppens et al., 2018 that shows the OMI-GOME2 AK and profiles), and can help the reader understand the added information content from the satellite measurements. For example, in Figure 2, is the largest improvement when you use the assimilated profiles at altitudes where the AKs peak? This can be added in the methodology section when you present GOME2 and OMI.

- What is the added value of using GOME2 + OMI versus using one or the other separately? In other words did you try a simulation with GOME2 alone and with OMI alone and see the effect on the comparison with sondes?

- The discussion in section 4 comes very late as the info (in particular on TM5 chemistry and MSR need to be mentioned earlier.

Detailed comments:

Page 2 L1: you can add also the NOx contribution to ozone formation

Page 2 L10-L15: the 0-6 km column is also chosen because satellite measurements are not very sensitive close to the surface. So the 0-6 km column has been (historically) chosen as a compromise for a "tropospheric column" that has some DOFS$\sim$1 (although many times less than that). If you have the AK plotted, you can see that.

P2 L28: add reference for IASI ozone (Boynard et al., 2018).

P2 L35: "Since UV-VIS" instruments are not very sensitive to the height of tropospheric ozone": this sentence has no meaning, please rephrase.

P3 line 7: remind the reader here one more time that the FAT column is [0-6km]

P3 L9: you introduce MSR without further info. What is MSR and why are you using it in particular; Please move P4 L4-5 here.

P3 L27: correlation in the ozone distribution... Correlation with what?

P4 L 12-14: higher ozone is "a general artifact of parametrization" what does this even mean? "without any further constraints to the model" makes also no sense. Can you please explain why the model shows higher ozone in the lower troposphere. Are we overestimating precursor emissions? Is photochemistry to blame? While you cite the chemistry scheme authors (Cariolle), you can mention here how the free model simulation was previously (if any) validated, especially for ozone.

P4 L29: (same as before): were these model "artefacts" seen in other publications, were they discussed before? Any suggested reason for their source (definition of tropopause etc.)

P4 L33: "with the exception of the UTLS (around 15 km)..": Why? Maybe the AK can help you?

P5/Fig1: since the current figure already occupies the whole page, you can add a difference plot so we see the clear contribution of the assimilated profiles.

P6 L5: the error bars on your figure are quite large and you attempt to show each station contribution in Fig 4 so I suggest to move Fig 4 and make it Fig 3. Unfortunately you don't tell us why for example in the northern hemisphere the assimilated O3 has smaller rms. Please attempt a more in depth discussion.

(current) Fig3: NP and NML are indistinguishable in color (pink and red), make it orange? Same applies to other figures

Again Figure 5: need to discuss the figure more and put it in context with the previous figures. I don't understand why the largest differences are in winter. I think you should present this by bands of latitude to understand where it is coming from. Please discuss more the reasons behind the assimilated O3 performance.

Figure 6 can be moved to before (after the discussion of Fig 2, or can be put in supplementary materials).

Grammatical/other minor edits:

Page 1: Abstract: change opening sentence to: "we derived global tropospheric ozone (O3) columns from assimilated O3 profiles of GOME-2A and OMI into the TM5 global chemistry transport model.

P1 L3: The horizontal model resolution is increased by a factor of six for more accurate results. To reduce...

P1 L6: assimilate->assimilated ozone fields

P1 L9: it turned out that -> Our results show that the residual method has large variations...

P2 L7,8,9 are not relevant to your study, they can be removed.

P 3 L2: [...] averaging kernels and the chemical ->averaging kernels in a chemical

P4 L1: change to: TM5 was used in two runs: a free model ....

P4 L18: and anthropogenic *precursors* emissions

P4 L20: NO2->NOx

P6 L7: ozone sonde station *location*.

Please read carefully the rest of the paper for other mistakes...

References :

Boynard, A., Hurtmans, D., Garane, K., Goutail, F., Hadji-Lazaro, J., Koukouli, M. E., Wespes, C., Vigouroux, C., Keppens, A., Pommereau, J.-P., Pazmino, A., Balis, D., Loyola, D., Valks, P., Sussmann, R., Smale, D., Coheur, P.-F., and Clerbaux, C.: Validation of the IASI FORLI/EUMETSAT ozone products using satellite (GOME-2), ground-based (Brewer–Dobson, SAOZ, FTIR) and ozonesonde measurements, Atmos. Meas. Tech., 11, 5125-5152, https://doi.org/10.5194/amt-11-5125-2018, 2018.

Keppens, A., Lambert, J.-C., Granville, J., Hubert, D., Verhoelst, T., Compernolle, S., Latter, B., Kerridge, B., Siddans, R., Boynard, A., Hadji-Lazaro, J., Clerbaux, C., Wespes, C., Hurtmans, D. R., Coheur, P.-F., van Peet, J. C. A., van der A, R. J., Garane, K., Koukouli, M. E., Balis, D. S., Delcloo, A., Kivi, R., Stübi, R., Godin-Beekmann, S., Van Roozendael, M., and Zehner, C.: Quality assessment of the Ozone_cci Climate Research Data Package (release 2017) – Part 2: Ground-based validation of nadir ozone profile data products, Atmos. Meas. Tech., 11, 3769-3800, https://doi.org/10.5194/amt-11-3769-2018, 2018.

---

## Referee Comment (RC2) · Anonymous Referee #2 · 4 Mar 2019

This is an interesting paper on determining tropospheric ozone from the assimilation of satellite measurements. The assimilation of satellite measurements is an important modern concept in determining global daily fields of trace constituents, one of those being tropospheric ozone. I suggest publication of this paper with mostly minor comments/changes listed below.

There are two main points to this paper and they are (1) testing TM5 assimilation versus TM5 free-running simulation, and (2) testing the quality of measuring 0-6 km column ozone from direct vertical integration, versus a "residual" difference from independent product/assimilation. Neither of these are particularly strong science points by themselves, but certainly fine to include in this study.

Two TM5 CTM simulations are compared, one simulation including assimilated METOP

[Figure]

GOME-2 and Aura OMI satellite measurements, and the other simulation being free-running that does not include assimilation of these measurements. A conclusion made, perhaps not all that surprising, is that the TM5 assimilated run which includes satellite measurements does better than the free-running TM5 model run. A question is why would we expect that a free-running simulation could compare better with ozonesondes than using TM5 with assimilated measurements?

Part of the analysis also involves comparing integrated ground-to-6 km column ozone (FAT column) from TM5 with a "residual" ground-to-6 km ozone calculated by differencing MSR total ozone minus TM5 modeled ozone column (from 6 km to top of atmosphere). If I understand correctly, the MSR total ozone is an assimilated data product for 1970-2017 that is independent of the TM5 modeled/assimilated ozone fields in the current study. MSR total ozone is derived from a composite of several satellite ozone measurements. (Line 4 page 4 actually says "from all available satellite measurements".)

The 0-6 km residual column ozone for comparison is derived by taking MSR DOAS total ozone and subtracting from this UV/VIS assimilated column ozone lying above 6 km. These are two very large column measurements

Differencing two large and independent column ozone measurements will be noisy due to basic statistics involving their inherent precision and accuracy numbers. It is not surprising that directly integrated 0-6 km FAT column ozone will be less noisy and will compare better with the ozonesondes. Are there any estimates for accuracy and precision for these two large independent columns that can be stated in the paper, or maybe better yet, an estimate of the accuracy and precision of the final derived 0-6 km residual ozone from their differences?

Also, directly integrated 0-6 km FAT column ozone ingests satellite UV/VIS measurements in the assimilation run; these satellite measurements have largely reduced sensitivity in detecting column ozone in the 0-6 km low-troposphere, perhaps up to 40%

or more. The DOF of ~1.2 in the tropics for GOME (line 14, page 9) as stated is a very good number, but it doesn't indicate how much real ozone variability is lost below 6 km due to insensitivity to ozone in the satellite retrievals. It may be that a lot or most of the 0-6 km column amount is coming from CTM modeling and retrieval climatology (for the UV/VIS assimilation run) rather than real measurement. It would be important to at least give some numbers for OMI and GOME-2 retrieval sensitivity for directly integrated 0-6 km column ozone, even if only qualitative.

SPECIFIC MINOR COMMENTS:

Line 6 in Abstract: "assimilated"

Paragraph starting Line 26, page 2: Should include reference(s) for TES/IASI IR retrieved ozone.

In the paper I'm assuming that RMS refers to standard deviation everywhere.

In Figure 2 and Figure 3, how actually are the uncertainty bars calculated? Are they all plotted as +/- one-sigma?

I was hoping that there were some more FAT comparisons with ozonesondes in the Figure 6 time series. The current Figure 6 compares only three sites, at latitude -70.7 deg, +19.4 deg, and +60.1 deg, but nothing for mid-latitudes. Including a couple of sites in mid-latitudes would help this figure and conclusions.

There are several typos in the manuscript that the authors may have already found and corrected.

---

## Author Comment (AC1) · 23 May 2019

**Answers to the referee comments in the interactive discussion regarding the article "Deriving tropospheric ozone from assimilated profiles"**

The article with all changes marked using the latexdiff program is included on page 8 of the current document.

All figures have been redrawn, but only significant changes will be addressed explicitly in the answers below.

**Referee #1**

1. *My main comments for the paper are that it lacks context; that is, it is not well discussed in context of what is done so far for ozone assimilation and how this work fits in the bigger picture.*
   A paragraph was added to the introduction which provides background information and references to other ozone assimilation work.

2. *The averaging kernels of OMI and GOME2-A can help you understand and analyze your results (or at least a discussion of AKs e.g. the recent publication by Keppens et al., 2018 that shows the OMI-GOME2 AK and profiles), and can help the reader understand the added information content from the satellite measurements. For example, in Figure 2, is the largest improvement when you use the assimilated profiles at altitudes where the AKs peak? This can be added in the methodology section when you present GOME2 and OMI.*
   A description of the AK, the retrieval sensitivity and two new references have been added to the methodology section. A new figure and explanatory text have been added to the results section. For this new figure we averaged the AKs of all retrievals that were assimilated into TM5. This figure shows that the largest improvement in Figure 2 is indeed where the AKs peak.

3. *What is the added value of using GOME2 + OMI versus using one or the other separately? In other words did you try a simulation with GOME2 alone and with OMI alone and see the effect on the comparison with sondes?*
   The added value of the combined assimilation of GOME-2 and OMI has been demonstrated in van Peet et al. (2018). In that paper, we did three assimilation runs: GOME-2 only, OMI only and GOME-2 + OMI as the referee suggests. Since the combined assimilation showed clear improvements over the assimilation of the separate instruments, we decided not

to repeat the single GOME-2 and OMI assimilation experiments for the current research.

4. *The discussion in section 4 comes very late as the info (in particular on TM5 chemistry and MSR need to be mentioned earlier.*
The information on TM5 chemistry has been copied to the methodology section.

5. *Page 2 L1: you can add also the NOx contribution to ozone formation*
Done

6. *Page 2 L10-L15: the 0–6 km column is also chosen because satellite measurements are not very sensitive close to the surface. So the 0–6 km column has been (historically) chosen as a compromise for a "tropospheric column" that has some DOFS~1 (although many times less than that). If you have the AK plotted, you can see that.*
The mean AKs of all assimilated GOME-2 and OMI retrievals have been plotted in the new figure 4 in the results section and a reference to that figure has been added to the introduction.

7. *P2 L28: add reference for IASI ozone (Boynard et al., 2018).*
The reference has been added.

8. *P2 L35: "Since UV-VIS instruments are not very sensitive to the height of tropospheric ozone": this sentence has no meaning, please rephrase.*
This sentence has been rephrased

9. *P3 line 7: remind the reader here one more time that the FAT column is [0-6km]*
The text has been updated.

10. *P3 L9: you introduce MSR without further info. What is MSR and why are you using it in particular; Please move P4 L4-5 here.*
The text has been moved.

11. *P3 L27: correlation in the ozone distribution... Correlation with what?*
We meant the spatial correlation between any two points in the ozone field. The text has been updated.

12. *P4 L 12-14: higher ozone is "a general artifact of parametrization" what does this even mean? "without any further constraints to the model" makes also no sense. Can you please explain why the model shows higher ozone in the lower troposphere. Are we overestimating precursor emissions? Is photochemistry to blame? While you cite the chemistry scheme authors (Cariolle), you can mention here how the free model simulation was previously (if any) validated, especially for ozone.*
The text at the start of section 3 has been clarified. Since the TM5 model

used in this research only uses the parameterized chemistry, there is no overestimating of precursor emissions or issues with the photo chemistry. The free model has been validated against sondes (see figure 2).

13. *P4 L29: (same as before): were these model "artefacts" seen in other publications, were they discussed before? Any suggested reason for their source (definition of tropopause etc.)*
    The option of a model artefact was a private communication by M. van Weele, during a discussion of these results. The source of the artefact is unknown.

14. *P4 L33:"with the exception of the UTLS (around 15 km)..": Why? Maybe the AK can help you?*
    We think that the sharp ozone gradients in this altitude range are not captured fully by the retrievals. Mean AKs for both GOME-2 and OMI have been added as a new plot in the results section (new figure 3).

15. *P5/Fig1: since the current figure already occupies the whole page, you can add a difference plot so we see the clear contribution of the assimilated profiles.*
    The difference plot has been added to the figure and the caption has been updated.

16. *P6 L5: the error bars on your figure are quite large and you attempt to show each station contribution in Fig 4 so I suggest to move Fig 4 and make it Fig 3. Unfortunately you don't tell us why for example in the northern hemisphere the assimilated O3 has smaller rms. Please attempt a more in depth discussion.*
    The error bars on P6, L5 refer to Figure 2: the comparison of ozone profiles from the assimilation with sondes. Old figures 3 and 4 (new 5 and 6) refer to the tropospheric columns. The same coloring of the sonde stations has been applied to both old figures 2 and 3, so we believe that for easier comparison they should be close together. We therefore left the order as it was.
    The tropical stations in old figure 4 (new figure 5) show larger rms and mean values than the extratropical stations. The reason for that is unknown, but it might be due to a higher variation in the sonde measurements.

17. *(current) Fig3: NP and NML are indistinguishable in color (pink and red), make it orange? Same applies to other figures*
    The colors and marker sizes for both figure 2 and (old) figure 3 have been harmonized which hopefully increases readability.

18. *Again Figure 5: need to discuss the figure more and put it in context*

*with the previous figures. I don't understand why the largest differences are in winter. I think you should present this by bands of latitude to understand where it is coming from. Please discuss more the reasons behind the assimilated O3 performance.*

After discussing the results on a station basis, we wanted to study the temporal variation throughout the year, as mentioned on page 8 (line 3) of the discussion paper. We discuss the temporal variation on page 8 (line 3–11) and in the discussion section on pages 11 (lines 34–35) and 12 (lines 1–5). We don't understand why the largest differences are in winter either, but it should be noted that the differences are only in the order of 2-3 DU, which are close to the uncertainty. The results are now presented by latitude bands and the discussion of the plot has been updated.

19. *Figure 6 can be moved to before (after the discussion of Fig 2, or can be put in supplementary materials).*

    The time series in figure 6 are meant as an illustration of the tropospheric columns at a few selected stations. To understand the variability of the results, it is important to first discuss the results in global terms before looking at specific stations. We therefore propose to leave the figure at its current location. On request of Referee #2 (see item 9), we've added more of these time series as supplementary material online.

20. *Page 1: Abstract: change opening sentence to: "we derived global tropospheric ozone (O3) columns from assimilated O3 profiles of GOME-2A and OMI into the TM5 global chemistry transport model."*

    The opening sentence has been updated.

21. *P1 L3: The horizontal model resolution is increased by a factor of six for more accurate results. To reduce...*

    Done

22. *P1 L6: assimilate → assimilated ozone fields*

    Done

23. *P1 L9: it turned out that → Our results show that the residual method has large variations...*

    Done

24. *P2 L7,8,9 are not relevant to your study, they can be removed.*

    Lines removed.

25. *P 3 L2: ... averaging kernels and the chemical → averaging kernels in a chemical*

    Data assimilation combines information from retrievals and model, so we believe that the current phrasing is correct.

26. *P4 L1: change to: TM5 was used in two runs: a free model*
    Changed

27. *P4 L18: and anthropogenic \*precursors\* emissions*
    Added precursor to the text

28. *P4 L20: NO2 → NOx*
    Changed

29. *P6 L7: ozone sonde station \*location\*.*
    Added "location" to the text.

30. *Please read carefully the rest of the paper for other mistakes...*
    We have checked the paper for other typos / mistakes.

**Referee #2**

1. *A question is why would we expect that a free-running simulation could compare better with ozonesondes than using TM5 with assimilated measurements?*
   We don't expect that a free-running simulation would compare better with sondes. It's included in the analysis for two reasons. First, it enables us to assess and quantify the effect of the assimilation. For example, if a feature in the ozone distribution is present in both free model run and assimilated ozone fields, it is not a result of the assimilation. Second, it is also a check if the assimilation algorithm is implemented correctly: if the model run would have performed better, there would be serious issues with the algorithm, the model or the data being assimilated.

2. *Are there any estimates for accuracy and precision for these two large independent columns that can be stated in the paper, or maybe better yet, an estimate of the accuracy and precision of the final derived 0–6 km residual ozone from their differences?*
   In the assimilation, the spatial correlation is assumed to be constant, and the advection scheme works on the errors only. The correlation is derived by the same method as described in van Peet et al. (2018). The errors on the profile are part of the assimilation output, and combining these two quantities allows the reconstruction of the covariance matrix in the vertical direction. The tropospheric part of this covariance matrix can be used to derive an error on the tropospheric column. The results of

this calculation is a map showing the relative error on the tropospheric FAT column and is added as a new plot to the article. The yearly mean relative FAT error varies between 11 and 18 %, depending on the location on Earth.

3. *It may be that a lot or most of the 0-6 km column amount is coming from CTM modeling and retrieval climatology (for the UV/VIS assimilation run) rather than real measurement.*
   To asses the amount of ozone coming from the CTM modelling is part of the reason to perform a free model run. See also the first comment by the referee above.

4. *It would be important to at least give some numbers for OMI and GOME-2 retrieval sensitivity for directly integrated 0–6 km column ozone, even if only qualitative.*
   GOME-2 and OMI are most sensitive in the altitude region between 100 and 5 hPa. To demonstrate this, the mean AK for all assimilated GOME-2 and OMI profiles has been calculated and plotted in a new figure (caption: Mean AKs calculated...). See also the answer to Referee #1, item 2.

5. *Line 6 in Abstract: "assimilated"*
   Corrected.

6. *Paragraph starting Line 26, page 2: Should include reference(s) for TES/IASI IR retrieved ozone.*
   Added reference to Boynard et al. 2018. See also Referee #1, item 7.

7. *In the paper I'm assuming that RMS refers to standard deviation everywhere.*
   The standard deviation is calculated according to

   $$s = \sqrt{\frac{1}{n-1} \times \sum \left(x_i - x_{\mathrm{mean}}\right)^2} \tag{1}$$

   When we're referring to the RMS, such as in the old figure 3 (new: figure 5), we use

   $$\mathrm{rms} = \sqrt{\frac{1}{n} \times \sum \left(x_i - y_i\right)^2} \tag{2}$$

   The difference is that the standard deviation is calculated for a single dataset, but the rms can be calculated between two different datasets.

8. *In Figure 2 and Figure 3, how actually are the uncertainty bars calculated? Are they all plotted as +/- one-sigma?*
   The error bars in both figures give the 25–75 percentile range. In figure

3 this was already mentioned in the caption. This information has been added to the caption of figure 2.

9. *I was hoping that there were some more FAT comparisons with ozonesondes in the Figure 6 time series. The current Figure 6 compares only three sites, at latitude -70.7 deg, +19.4 deg, and +60.1 deg, but nothing for mid-latitudes. Including a couple of sites in mid-latitudes would help this figure and conclusions.*
The time series for all ozone sonde stations used in the analysis with more than 10 collocations are now available in the supplementary material online. A line referring the reader to the supplementary material has been added to the paper.

10. *There are several typos in the manuscript that the authors may have already found and corrected.*
We have checked the paper for other typos / mistakes.

**References**

[revised manuscript text omitted]